# Gut Microbiota and Acute Diverticulitis: Role of Probiotics in Management of This Delicate Pathophysiological Balance

**DOI:** 10.3390/jpm11040298

**Published:** 2021-04-14

**Authors:** Andrea Piccioni, Laura Franza, Mattia Brigida, Christian Zanza, Enrico Torelli, Martina Petrucci, Rebecca Nicolò, Marcello Covino, Marcello Candelli, Angela Saviano, Veronica Ojetti, Francesco Franceschi

**Affiliations:** 1Emergency Medicine, Fondazione Policlinico Universitario A. Gemelli IRCCS, 1-00168 Rome, Italy; marcello.covino@policlinicogemelli.it (M.C.); marcello.candelli@policlinicogemelli.it (M.C.); veronica.ojetti@unicatt.it (V.O.); francesco.franceschi@unicatt.it (F.F.); 2Università Cattolica del Sacro Cuore, 1-00168 Rome, Italy; laura.franza@policlicnigemelli.it (L.F.); christian.zanza@live.it (C.Z.); rikho88@gmail.com (E.T.); martina.petrucci@policlinicogemelli.it (M.P.); rebecca.nicolo@policlinicogemelli.it (R.N.); saviange@libero.it (A.S.); 3Unit of Gastroenterology, Department of Systems Medicine, Tor Vergata University, 2-00133 Rome, Italy; mattiabrigida@hotmail.it

**Keywords:** gut microbiota, probiotics and gut disease, probiotics and acute diverticulitis, probiotics and diverticular disease, probiotics mechanism of action

## Abstract

How can the knowledge of probiotics and their mechanisms of action be translated into clinical practice when treating patients with diverticular disease and acute diverticulitis? Changes in microbiota composition have been observed in patients who were developing acute diverticulitis, with a reduction of taxa with anti-inflammatory activity, such as Clostridium cluster IV, Lactobacilli and Bacteroides. Recent observations supported that a dysbiosis characterised by decreased presence of anti-inflammatory bacterial species might be linked to mucosal inflammation, and a vicious cycle results from a mucosal inflammation driving dysbiosis at the same time. An alteration in gut microbiota can lead to an altered activation of nerve fibres, and subsequent neuronal and muscular dysfunction, thus favoring abdominal symptoms’ development. The possible role of dysbiosis and mucosal inflammation in leading to dysmotility is linked, in turn, to bacterial translocation from the lumen of the diverticulum to perivisceral area. There, a possible activation of Toll-like receptors has been described, with a subsequent inflammatory reaction at the level of the perivisceral tissues. Being aware that bacterial colonisation of diverticula is involved in the pathogenesis of acute diverticulitis, the rationale for the potential role of probiotics in the treatment of this disease becomes clearer. For this review, articles were identified using the electronic PubMed database through a comprehensive search conducted by combining key terms such as “gut microbiota”, “probiotics and gut disease”, “probiotics and acute diverticulitis”, “probiotics and diverticular disease”, “probiotics mechanism of action”. However, the amount of data present on this matter is not sufficient to draw robust conclusions on the efficacy of probiotics for symptoms’ management in diverticular disease.

## 1. Introduction—Microbiota in Health and Disease

Over the last three decades, the importance of gut microbiota in determining health and disease has become increasingly clear.

Already in the late 1800s, researchers were warning the public that the bacteria living in our intestine could be “pathological” [1] even though the concept of dysbiosis had yet to be formulated.

Microbiota has proven to be an important player in the pathogenesis of many different diseases, ranging from more “obvious” disorders, for instance small intestine bacterial overgrowth (SIBO) [2], to more complicated diseases, such as immune deficits, thyroid disorders, neurodegenerative diseases [3,4,5], and have also been linked to mood disorders [6].

In this review, our aim was to discuss the potential role of probiotics for the treatment of diverticular disease and acute diverticulitis, with particular attention to their possible mechanisms of action.

The most important mechanisms through which microbiota can influence systemic health is through immune/inflammatory mechanisms [7,8]. The presence of certain bacterial strains can exert regulatory functions, improving immune-tolerance and stimulating regulatory T-cell (T-reg) expression [9]. This activity was observed, for instance, in *Bacteroides fragilis* [10] and in some Clostridium species [11].

Immune modulation also takes place through the production of short-chain fatty acids (SCFAs). Indeed, when digesting fibre, bacteria produce a vast array of SCFAs. Butyrate is particularly important among them, as it directly modulates the expression of histone deacetylase (HDAC), consequently increasing the expression of T-regs. Moreover, its role in obesity and metabolic control is not yet clear [12].

Additionally, a “healthy” microbiota works as a physical barrier against pathogens, stopping them from overcoming the gut mucosa and spreading systemically [13]. Indeed, beneficial species compete for nutrients and can produce antimicrobial substances which do not allow the growth of other microorganisms. Yet, it is worth noting that a certain microbial population can shift from protective to harmful even in the same individual based on circumstance.

In Table 1 there is a short summary of the relevant literature we discussed.

## 2. Materials and Methods

For this review, articles were identified using the electronic PubMed database through a comprehensive search conducted by combining key terms such as “probiotics”, “gut microbiota”, “probiotics and gut disease”, “probiotics and acute diverticulitis”, “probiotics and diverticular disease”, “probiotics mechanism of action”, “gut immunology and probiotics”. English-language articles were screened for relevance. A full review of publications for the relevant studies was conducted, including additional publications that were identified from individual article reference lists. At first, the literature search was conducted individually by the single authors, who then compared their results, to include in the review only the most recent and relevant papers.

## 3. What Is a Healthy Microbiota?

As discussed above, the role of microbiota in determining health and disease, via immune modulation, is becoming increasingly clear. Some microbes have been identified as definitely pathogenic, for instance *Clostridium difficile*, yet there still are some grey areas, in particular when it comes to determine what defines a healthy microbiota [14].

An example of this comes from studies on microbiota and cancer: *Helicobacter pylori* is a known risk factor for the development of gastric cancer, but it has a protective effect against oesophageal cancer [15]. Similarly, *Escherichia coli* protects against pancreatic cancer, but favours colorectal and liver tumours [16]. In these cases, the ambivalent role of these bacteria was determined by the site of colonisation.

It is, indeed, important to underline that, even though it is common to refer to “gut microbiota”, this does change widely throughout the gastrointestinal tract. In the oral cavity, for instance, it is common to find *Neisseria spp.*, which is instead difficult to find in other sites [17]. The stomach has a completely different microbiota than all the other parts of the gastrointestinal tract in healthy patients, but resembles oesophageal or intestinal microbiota in those with gastric cancer [18]. The small intestine even has a different composition based on which tract is being studied, and its composition is different from that of the colon.

Even considering specific parts of the gastrointestinal tract, microbiota changes with age [19], and even depending on geographic localisation, diet and ethnicity [20].

Yet, even though it is not possible to precisely define what species compose a healthy microbiota, it is worth noting that some general characteristics have been observed: a healthy microbiota is made up of a dynamic and diverse community of microbes, which is able to self-regulate [21].

While these do seem to be vague concepts, it is interesting to notice that the modern, western diet directly affects microbiota diversity, reducing its capacity to bounce back when frankly pathogenic species start colonising the gut [22].

## 4. Gut Disease and Microbiota

While gut microbiota influences all aspects of human health, its action on the gastrointestinal system is particularly important.

Indeed, gut microbiota directly modulates gastrointestinal homeostasis, through a variety of mechanisms. The presence of pathogens at the intestinal barrier, for instance, can directly damage the gastrointestinal mucosa and determine inflammation. This is the mechanism through which *C. difficile, Salmonella spp* and others can create direct intestinal damage [23].

Direct damage also activates immunologic pathways, particularly through the activation of the inflammasome. IL-1β and IL-18 are a direct consequence of its activation, causing pyroptosis, a particular form of immune-mediated cellular death [24]. Meanwhile, the activation of the inflammasome is important to maintain gut homeostasis, as it helps restore microbiota eubiosis, and it can also lead to chronic gut inflammation, which in turn promotes an inflammatory gut microbiota, in a difficult-to-break vicious circle.

It was observed that *Bifidobacterium adolescentis, Lactobacillus, Phascolarctobacterium, Akkermansia muciniphila* are all reduced in patients with intestinal inflammation. Interestingly, when present, they are capable of reducing inflammation, particularly acting on C-reactive protein (CRP), IL-6, and tumour necrosis factor (TNF)-α [25]. The action on these inflammatory mediators partly explains how gut microbiota can influence systemic health.

In the gut, the action of these mediators has direct consequences in terms of permeability and inflammation, both extremely important in the pathogenesis of different gastrointestinal disorders [26]. The presence of dysbiosis can trigger the development of inflammatory bowel disease (IBD) and irritable bowel syndrome (IBS). The role of microbiota has been particularly underlined in the pathogenesis of IBDs, in which dysbiosis is marked by the presence of *Mycobacterium avium* subsp. *paratuberculosis, Fusobacterium nucleatum,* adherent–invasive *E. coli* [27].

Inflammation caused by microbiota dysbiosis is also responsible of liver disorders, particularly non-alcoholic liver steatosis. In this case, there is a direct colonisation of the bile ducts, which adds up to the action of bacterial metabolic products. Interestingly, the capacity of bacteria to metabolise biliary salts is also linked to dysbiosis [28].

Inflammation is also a well-known risk factor for the development of cancer, and it is no different in the intestine; the impact of microbiota composition in modulating the intestinal immunologic niche has proven essential in different forms of cancer [16].

## 5. Microbiota Modulation: The Case for Probiotics

Given the importance of microbiota in human health, the possibility of modulating it to obtain benefits is an interesting potential therapeutic target.

Microbiota modulation can take place in two different ways, either through the use of antibiotics or through the use of probiotics.

Antibiotic use can target specific pathological bacteria in the gut, eliminating it, as in the case of vancomycin in *C. difficile* infection [29]. Some antibiotics, such as rifaximin, can instead be used to target a larger number of pathogens, improving conditions such as IBS, SIBO and preventing encephalopathy in patients suffering from liver disease [30]. Ozone, for instance, also seems to have a similar capacity to reduce inflammation in the gut, also through microbiota modulation [31,32].

Yet the use of antibiotics can be tricky. The use of antibiotics can reduce taxonomic diversity of gut microbiota and induce resistance mechanisms in pathogenic species, while also favouring the development of *C. difficile* infection and other dangerous pathogens [33].

Using probiotics can provide similar positive effects to antibiotics, in reducing pathogens, but at the same time avoiding many of the side effects [34]. Probiotics have been used with different degrees of success in patients suffering from IBS, gastroenteritis and even in *C. difficile*-associated diarrhoea [35]. Probiotics have also been used in neonatal sepsis and necrotising enterocolitis [36].

Probiotic use has also been discussed in extra-gut conditions, such as autism and acute respiratory infections [37,38].

Some authors even suggest that microbiota modulation through probiotics could be beneficial in preventing cancer [39] and improving response to chemotherapy, reducing gastrointestinal side effects [40].

## 6. Microbiota and Acute Diverticulitis: What We Know, What We Can Do

Diverticulitis is usually merely considered an inflammation of a herniation of colonic mucosa and submucosa, through the muscle layer. While it is obvious that the inflammatory process is driven by the resident microbiota, it is not as clear whether the microbial population may play a role in promoting the herniation in the first place [41].

It is known that dietary factors are key in determining the onset of diverticular disorder and they also determine changes in microbiota composition. It is difficult to determine whether the changes in the composition of the microbiota may act as an enhancing factor in the development of diverticulitis, but it is worth noting that the microbial species associated with diverticular disease are *Enterobacteriaceae, Streptococcus* and *Bacteroides*, while reducing the expression of “good” bacteria, such *Bifidobacteria* and *Lactobacilli* [42].

The role of microbiota in determining diverticular inflammation is instead far clearer. Recurring diverticulitis not susceptible to surgery has, indeed, been treated with faecal transplant, leading to complete remission [43]. Additionally, it has been reported that a patient, with moderate diverticular disease, who underwent faecal transplant for *C. difficile* infection, developed her first ever episode of diverticulitis after the procedure, further proving that, while it may not take part in the first, more mechanical stages of the disease, microbiota is not a bystander during the inflammatory phase [44].

Changes in microbiota composition have been observed in patients who were developing acute diverticulitis, with a reduction of taxa with anti-inflammatory activity, such as *Clostridium* cluster IV, Lactobacilli and Bacteroides. At the same time, overgrowth of Bifidobacteria, Enterobacteriaceae and *Akkermansia* have been reported [41]. This diversity, also characterised by an increase in Proteobacteria, has proven significant and could be used for an early diagnosis of the disease [45].

Overall, the importance of microbiota in diverticular disease has been demonstrated indirectly by the therapies used to treat the disorder. Indeed, the use of rifaximin and probiotics has proven interesting, even though mostly through clinical trials.

Rifaximin has been used to treat Symptomatic Uncomplicated Diverticular Disease in a clinical trial and results were encouraging, not only in terms of symptom control but also as far as faecal microbiota composition, with a reduction of *Roseburia, Veillonella, Streptococcus* and *Haemophilus* [46]. In another study, there was also a reduction in *Akkermansia* [47].

Lactobacilli have been demonstrated to reduce Symptomatic Uncomplicated Diverticular Disease, particularly in reducing bloating and abdominal pain [48], while *Lactobacillus salivarius*, *Lactobacillus acidophilus* and *Bifidobacterium lactis* have proven effective in managing acute diverticulitis [49]. A double-blind randomised control trial was recently published by Ojetti et al., in which the authors tested the efficacy of *L. reuteri* 4659 in treating patients affected by acute uncomplicated diverticular diseases (AUD). Supplementation of the standard AUD therapy with this specific probiotic with an anti-inflammatory effect significantly reduced abdominal pain and inflammatory markers compared to those who were not taking the same supplementation [50]. These data were also observed in another paper, published in 2019, in which the authors demonstrated that in patients affected by AUD supplementation with a mix of probiotics *B. lactis* LA 304, *L. salivarius* LA302 and *L. acidophilus* LA 201, with a well-known capacity for reducing TNF levels, in combination with the standard antibiotic therapy reduced PCR level and the abdominal pain compared to patients with the disease who did not receive the supplementation [51].

In Figure 1, the described mechanisms are shown.

The immunomodulatory properties of probiotics are also associated with cytokine release, in particular tumour necrosis factors (TNF), transforming growth factor (TGF), interleukins (IL), interferons (IFN) and chemokines derived from immune cells that regulate the innate and adaptive compartments of the immune system. Some cell wall components of Lactobacilli and Bifidobacteria, such as lipoteichoic acid, stimulate nitric oxide (NO) synthesis, which is key in pathogen-infected cell death mechanisms, induced by macrophages via TNF-α secretion. Furthermore, two surface receptors involved in phagocytosis, namely FcγRIII and toll-like receptors (TLR), are also upregulated by NO.

Probiotics have been proven to interact with enterocytes, Th1, Th2, Treg cells and dendritic cells in the gut and also regulate the adaptive immunity.

## 7. Probiotics in Diverticulitis: Mechanisms of Action

To better understand how probiotics exert an impact on diverticular disease and diverticulitis, attention should be focused on some aspects of pathophysiology linked to microbiota.

Recent observations support that a dysbiosis characterised by decreased presence of anti-inflammatory bacterial species might be linked to mucosal inflammation, and a vicious cycle results from a mucosal inflammation driving dysbiosis at the same time [51].

Another key element is the possible role of dysbiosis and mucosal inflammation in leading to dysmotility: an alteration in gut microbiota, indeed, can lead to altered nerve fibre activation and subsequent neuronal and muscular dysfunction, thus favouring the development of diverticulosis, while also possibly inducing abdominal symptoms [52]. Altered motility is linked, in turn, to bacterial translocation from the lumen of the diverticulum to perivisceral area. There, a possible activation of Toll-like receptors (TLR) [53] has been described, with a subsequent inflammatory reaction at the level of the perivisceral tissues [54].

Moreover, Foligne and colleagues [55] have studied, in the context of IBD, thirteen strains of probiotics in terms of anti-inflammatory properties and, among these, *L. acidophilus* and *L. salivarius* Ls33 seemed to be the best-performing concerning increased induction of IL-10 and decreased induction of IL-12.

Data coming from in vitro and in vivo studies, concerning *L. salivarius* Ls33, suggest that its administration is linked to an improved recovery of inflamed tissue in a rat colitis model [56].

This evidence has led researchers to consider changes in peri-diverticular bacterial flora as a critical element in acute diverticulitis pathogenesis, and a similar model was applied to explain acute appendicitis pathogenesis. Basically, stasis of faecal material within diverticula can be favoured by a prolonged colonic transit, which in turn predisposes to altered microflora and bacterial overgrowth. Mucosal barrier function can be consequently impaired and provoke an inflammatory reaction by means of cytokine release; a low-grade, localised microscopic colitis may result, which can evolve towards microperforation and show the clinical features of acute diverticulitis [57].

Being aware that bacterial colonisation of diverticula is involved in the pathogenesis of acute diverticulitis, the rationale for the potential role of probiotics in the treatment of this disease becomes clearer. As Quigley reports, probiotics may be able to modify the localised and persistent inflammation, present in some patients who are between acute bouts of diverticulitis. Acting on inflammation they may also act on symptom development, in individuals affected by uncomplicated diverticular disease [58].

In addition, the intestinal bacterial flora produces the outer membrane vesicles which play an important role in microbiota-host communication, the interaction of which takes place thanks to the action of adhesins, sulfatases and proteases and pathways such as micropinocytosis, clathrin- and lipid raft-dependent endocytosis [59].

These outer membrane vesicles have a positive impact on mucosal immunity and its signaling pathways. This would therefore seem to be another mechanism of action that can be exploited through the administration of probiotics in the context of various intestinal diseases, including diverticulitis. However, there is still scarcity of literature on the exact mechanisms of vesicles secreted by various species of microbiota and probiotics [57,60].

In a paper published by Brandimarte et al. [61], the pros and cons were discussed regarding evidence of probiotic action in diverticular disease. Recognising that overgrowth and alteration of gut microbiota can play a role in the development of inflammation related to diverticular disease, there is a clear rationale for administering probiotics with the aim to restore a healthy microenvironment in the colon. Different mechanisms have been discussed, such as a decrease in bacterial translocation, competitive inhibition of pathogenic and proinflammatory bacterial strains overgrowth, down-regulation of inflammatory cytokines, together with an improvement in mucosal defence, due to enhanced tight junction integrity [48,62,63].

However, the amount of data present on this matter is not sufficient to draw robust conclusions on the efficacy of probiotics for symptom management in diverticular disease, and this was confirmed by a recent expert consensus with a submaximal level of agreement [64].

Indeed, concerning therapy with probiotics, there are no established protocols defining which strain, what dosage and for how long to use them, and this reflects the absence of reliable meta-analyses in this regard [61].

Literature should be broadened as new mechanisms of action come to light from the many investigations being currently conducted in numerous centres around the globe, and new protocols should be established in order to study how exactly probiotic administration could make the difference in the management of diverticular disease and acute diverticulitis.

## Figures and Tables

**Figure 1 jpm-11-00298-f001:**
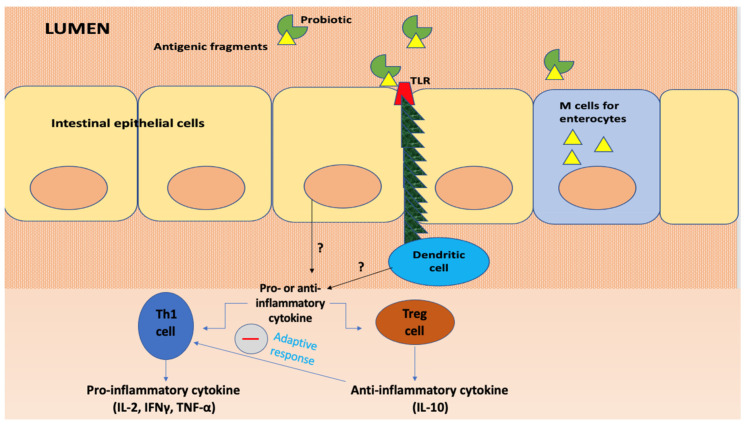
The immunomodulatory properties of probiotics.

**Table 1 jpm-11-00298-t001:** Summary of the most relevant research.

Title	Topic	Reference Number
Belkaid Y, Hand TW. “Role of the microbiota in immunity and inflammation”. Cell. 2014	Microbiota-driven immune responses	[8]
Iebba V, Totino V, Gagliardi A, Santangelo F, Cacciotti F, Trancassini M, et al. “Eubiosis and dysbiosis: the two sides of the microbiota”. New Microbiol. 2016	Dysbiosis and eubiosis in defining a “healthy” gut microbiota	[14]
Cianci R, Franza L. “The Interplay between Immunity and Microbiota at Intestinal Immunological Niche: The Case of Cancer”. 2019	The intestinal niche in health and disease	[16]
Lange K, Buerger M, Stallmach A, Bruns T. “Effects of Antibiotics on Gut Microbiota”. Dig Dis. 2016	The role of antibiotics in shaping the microbiota	[33]
Suez J, Zmora N, Segal E. “The pros, cons, and many unknowns of probiotics”. 2019	The role of probiotics in shaping the microbiota	[35]
Kim SK, Guevarra RB, Kim YT, Kwon J, Kim H, Cho JH, et al. “Role of Probiotics in Human Gut Microbiome-Associated Diseases”. J Microbiol Biotechnol. 2019	Probiotics in gut-microbiota associated diseases	[38]
Ticinesi A, Nouvenne A, Corrente V, Tana C, Di Mario F, Meschi T. “Diverticular Disease: a Gut Microbiota Perspective. Journal of gastrointestinal and liver diseases”. JGLD. 2019	The role of microbiota in diverticular disease	[41]
Ojetti V, Petruzziello C, Cardone S, Saviano L, Migneco A, Santarelli L, et al. “The Use of Probiotics in Different Phases of Diverticular Disease. Reviews on recent clinical trials”. 2018	Action of probiotics in diverticular disease	[49]
Petruzziello C, Migneco A, Cardone S, Covino M, Saviano A, Franceschi F, et al. Supplementation with Lactobacillus reuteri ATCC PTA 4659 in patients affected by acute uncomplicated diverticulitis: a randomised double-blind placebo controlled trial. 2019	Action of probiotics in diverticular disease	[50]
Petruzziello C, Marannino M, Migneco A, Brigida M, Saviano A, Piccioni A, et al. The efficacy of a mix of three probiotic strains in reducing abdominal pain and inflammatory biomarkers in acute uncomplicated diverticulitis. European review for medical and pharmacological sciences. 2019	Action of probiotics in diverticular disease	[51]

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
