# Peer review of "Gut Microbiota and Acute Diverticulitis: Role of Probiotics in Management of This Delicate Pathophysiological Balance"

_jpm, 2021, doi:10.3390/jpm11040298_

Round 1
Reviewer 1 Report
In this review article the authors have summarized the importance got microbiome and its role in normal functioning of gastrointestinal tract, more importantly in diverticular disease. The authors suggest that gut microbiome may play a role in the development of diverticulosis as well as diverticulitis.
The authors have made a good effort to provide a simplified explanation based on the available literature. There are several grammatical errors that needs correction. I have only mentioned a few here
Abstract: "from the lumen of the diverticulum to THE perivisceral area"
Section 7. thirteen strains of probiotics in terms of antiinflammatory properties" hyphen is missing for anti-inflammatory
Section 7. "These evidences have lead researchers to consider change"- it should be 'led'
and there are many such minor mistakes that need to be corrected by a first language English speaker.
Also in section 7 " An alteration in gut microbiota can lead to an altered nerve fibers’ activation and subsequent neuronal and muscular dysfunction, thus favoring abdominal symptoms’ development. (46)" this statement should say the change in microbiota can lead to development on diverticulosis and not abdominal symptoms.
Author Response
Reviewer 1
In this review article the authors have summarized the importance got microbiome and its role in normal functioning of gastrointestinal tract, more importantly in diverticular disease. The authors suggest that gut microbiome may play a role in the development of diverticulosis as well as diverticulitis.
The authors have made a good effort to provide a simplified explanation based on the available literature. There are several grammatical errors that needs correction. I have only mentioned a few here
Abstract: "from the lumen of the diverticulum to THE perivisceral area"
Section 7. thirteen strains of probiotics in terms of antiinflammatory properties" hyphen is missing for anti-inflammatory
Section 7. "These evidences have lead researchers to consider change"- it should be 'led'
and there are many such minor mistakes that need to be corrected by a first language English speaker.
Thank you for your comments. We have thoroughly revised our paper from a linguistic point of view
Also in section 7 " An alteration in gut microbiota can lead to an altered nerve fibers’ activation and subsequent neuronal and muscular dysfunction, thus favoring abdominal symptoms’ development. (46)" this statement should say the change in microbiota can lead to development on diverticulosis and not abdominal symptoms.
We corrected the sentence, highlighting the importance in the development od diverticulosis, while also, possibly, creating abdominal discomfort.
Reviewer 2 Report
The authors wrote a review about gut microbiota and its role in diverticular disease. Although the review is professionally written in a constructed manner. some concerns have to be addressed.
Major
- I feel the background contents were written in detail however, they are somehow long. There are a lot of sentences before the information about the diverticular disease. Please make it shorter if possible. Moreover, please consider making tables or figures to summarize the relevant research available. Readers may stop reading before reaching the main contents of this review.
- On page 3, ‘What is a healthy microbiota?’ section, the authors wrote ‘Helicobacter pylori is a known risk factor for the development of gastric cancer, but it has a protective effect against oesophageal cancer. The authors should add references to this sentence.
- Please consider making tables or figures to help readers understand the possible mechanism of action about related cytokines and receptors. If readers are not familiar with the microbiota and its molecular interactions, it is difficult for them to understand the contents.
Minor
- On page 5, the sentence ‘It is difficult to determine whether the changes in the composition of the microbiota my act as an enhancing factor in the development of diverticulitis, ‘ may be wrong. Does ‘my’ mean ‘may’?
- In the last part of page 5, is ‘nothing’ misspelling of ‘noting’?
- On page 6, the sentence ‘Proteobacteria, has proven significant and could be used for early early diagnosis of the disease. The ‘early’ is duplicated
- On page 7 does ‘microperforation’ mean actual very tiny perforation?
Author Response
I feel the background contents were written in detail however, they are somehow long. There are a lot of sentences before the information about the diverticular disease. Please make it shorter if possible. Moreover, please consider making tables or figures to summarize the relevant research available. Readers may stop reading before reaching the main contents of this review.
Thank you for your comments. We have summarized the contents of the background and added a short table, summarizing what we think are the most relevant papers.
On page 3, ‘What is a healthy microbiota?’ section, the authors wrote ‘Helicobacter pylori is a known risk factor for the development of gastric cancer, but it has a protective effect against oesophageal cancer. The authors should add references to this sentence.
We have added a citation on the role of H. pylori in oesophageal cancer.
Please consider making tables or figures to help readers understand the possible mechanism of action about related cytokines and receptors. If readers are not familiar with the microbiota and its molecular interactions, it is difficult for them to understand the contents.
We have added a figure, in which the role of cytokines is summarized.
Minor
On page 5, the sentence ‘It is difficult to determine whether the changes in the composition of the microbiota my act as an enhancing factor in the development of diverticulitis, ‘ may be wrong. Does ‘my’ mean ‘may’?
In the last part of page 5, is ‘nothing’ misspelling of ‘noting’?
On page 6, the sentence ‘Proteobacteria, has proven significant and could be used for early early diagnosis of the disease. The ‘early’ is duplicated
Thank you for finding these typos we had not noticed.
On page 7 does ‘microperforation’ mean actual very tiny perforation?
Yes, it does.